# Joint Angle and Frequency Estimation Using One-Bit Measurements

**DOI:** 10.3390/s19245422

**Published:** 2019-12-09

**Authors:** Zeyang Li, Junpeng Shi, Xinhai Wang, Fangqing Wen

**Affiliations:** 1National Demonstration Center for Experimental Electrical and Electronic Education, Yangtze University, Jingzhou 434023, China; 201700742@yangtzeu.edu.cn; 2State Key Laboratory of Marine Resource Utilization in South China Sea, Hainan University, Haikou 570228, China; 3National University of Defense Technology, Hefei 230037, China; 15667081720@163.com; 4Nanjing Marine Radar Institute, Nanjing 211153, China

**Keywords:** one-bit quantification, angle and frequency estimation, array signal processing, parallel factor analysis

## Abstract

Joint angle and frequency estimation is an important branch in array signal processing with numerous applications in radar, sonar, wireless communications, etc. Extensive attention has been paid and numerous algorithms have been developed. However, existing algorithms rely on accurately quantified measurements. In this paper, we stress the problem of angle and frequency estimation for sensor arrays using one-bit measurements. The relationship between the covariance matrices of one-bit measurement and that of the accurately quantified measurement is extended to the tensor domain. Moreover, a one-bit parallel factor analysis (PARAFAC) estimator is proposed. The simulation results show that the angle and frequency estimation can be quickly achieved and correctly paired.

## 1. Introduction

Source localization is one of the most important branches of array signal processing [1,2]. It has been actively conducted in the fields of communication, radar, sonar, seismic exploration, and cognitive radio [3,4,5,6,7]. Source localization using a sensor array always involves spectrum estimation, such as direction-of-arrival (DOA) estimation, delay estimation, frequency estimation, polarization estimation, or a combination of them. Among the enormous research topics, joint angle and frequency estimation are particularly prominent, since the two parameters are very important in various fields, and they can improve the detection ability and anti-interference ability of the spatial source signals. For instance, the two parameters can be adopted in passive radar systems for target locating and tracking; in space division multiple access-based wireless communications systems, these two parameters can be utilized to locate the user and allocate pilot tones; also, these two parameters are useful for channel estimation and beamforming. Moreover, the algorithms used for joint DOA and frequency estimation can be easily extended for angle-delay estimation, delay-frequency estimation, angle-delay-frequency estimation, etc., since problems of multiple parameter estimation using a sensor array are very similar to each other. Due to the above reasons, we focus on the problem of joint DOA and frequency estimation in this paper.

In the past decades, various spectrum estimation algorithms have been proposed. Typical algorithms including multiple signal classification (MUSIC) [8,9,10], estimating signal parameters via rotational invariance technique (ESPRIT) [11,12], propagator method [13], maximum likelihood (ML) [14,15], tensor-based approaches [16,17,18,19,20], and optimization-aware algorithms [21,22,23,24,25,26,27]. Generally speaking, MUSIC is computationally inefficient as it requires multiple peak search. Also, an ML estimator has high complexity due to exhaustive iteration. Unlike MUSIC and ML, ESPRIT can obtain closed-form solution, at the expense of decreased array aperture. Both MUSIC and ESPRIT need eigen decomposition to obtain the signal subspace or the noise subspace, and the complexity of eigen decomposition is on the third order of the matrix dimension. To avoid eigen decomposition, the propagator method has been introduced, which can obtain the subspaces via the least squares (LS) method. Optimization-aware algorithms always make sure accurate spectrum estimation; however, they are often too complex to be applied. Usually, the above-mentioned algorithm rely on matrix decomposition—approaches based on tensor decomposition are often superior than the matrix-based methods as they have better de-noising performance than the latter. A tensor can be viewed as a multi-dimensional (more than three) vector (matrix can be interpreted as a two-dimensional vector). Two tensor models are frequently used, namely parallel factor analysis (PARAFAC) and Tucker tensor. The former factorize a high-dimensional low-rank tensor into sums of rank-one tensors, the latter is highly analogy to multi-dimensional eigen decomposition.

To perform high-resolution spectrum estimation, some optional techniques are helpful, e.g., large-scaler sensor array, wideband signals, high-precision, analog-to-digital converter (ADC). However, these techniques would bring massive measurements and challenge the sampling system. On the one hand, high-precision ADCs are expensive and are of high energy consumption, which leads to an increased cost and complexity. On the other hand, massive measurements require more storage and processing resource; thus the real-time performance of the system is difficult to guarantee. To overcome these disadvantages, the concept of compressed sensing (CS) has been proposed [28,29], which may provide new inspirations for signal acquisition and processing. Several CS-based sampling frameworks have been developed, e.g., random sampler, random demodulation, modulated wideband converter, time encoding machine. Nevertheless, these architectures rely on high-precision quantification. Recently, the concept of one-bit CS [30], i.e., quantization with only one bit, has been proposed. With one-bit quantization, only one data bit need to be stored and processed, the system complexity can be reduced accordingly. Owing to its potential prospect, one-bit quantization has aroused much attentions in massive multiple-input multiple-output (MIMO) communications and DOA estimation [31]. Generally, spectrum estimation with one-bit measurements is linked to a sparse inverse problem, which is resolved via the optimization method. More recently, a one-bit MUSIC framework was driven in [32]. It is dproven that the covariance matrix with one-bit measurements can be approximated by a scaled unquantized covariance matrix, thus the traditional subspace algorithms can be directly applied. Besides, many efforts have been devoted to the sparse recovery problem from one-bit measurement [33,34]. In addition, some works have been done to the waveform design problems in MIMO radar with the problem of one-bit DAC [35].

It should be pointed out that, as mentioned previously, one-bit quantization is a lossy compression method; thus, the performance of spectrum estimation algorithms with one-bit quantization are suffering from degradation. For performance enhancement, this paper tries to integrate the tensor approach with joint DOA and frequency estimation in the presence of one-bit measurement. To this end, a new spatial-time sampling framework is presented, in which a one-bit ADCs are adopted. The relationship between the covariance matrices of one-bit measurement and that of the unquantized covariance matrix is extended to the tensor domain. A one-bit PARAFAC algorithm is proposed for joint DOA and frequency estimation, in which closed-form and automatically paired parameters are achieved. Compared with the one-bit ESPRIT algorithm, the proposed algorithm offers a more accurate estimation performance. Numerical simulations verify the effectiveness of the proposed algorithm.

The rest of the paper is organized as follows. In Section 2, we present the signal model and analyze the impact of measurement process on signal noise. Section 3 provides the proposed ASCS scheme. The simulation results are given in Section 4. Finally, conclusions are given in Section 5.

Notation: Lower case and capital letters in bold denote, respectively, vectors and matrices. The superscript ${\left(•\right)}^{T}$, ${\left(•\right)}^{H}$, ${\left(•\right)}^{-1}$, and ${\left(•\right)}^{†}$ represent the operators of transpose, Hermitian transpose, inverse, and pseudo-inverse, respectively; ${\left\|•\right\|}_{F}$ denotes the Frobenius norm. angle(·) returns the phase of a vector in radian. E[⋅] is to get the mathematical expectation of a variable. $D_{m}\left(A\right)$ returns a diagonal matrix with the diagonal entities are the m-th row of A.

## 2. Signal Model and One-Bit Quantization

Herein, we consider a uniform linear array scenario with *M*-element sensors, the inter-element interval is *d*. Suppose that there are *K* uncorrelated narrow-band sources appearing on the far-field of the array. Besides, the signal and noise are uncorrelated, and both of them are modeled as independent, zero-mean, circular, complex Gaussian random processes, then the received signal of the m-th sensor can be expressed as [36]
(1)$$y_{m}\left(t\right)=\sum_{k=1}^{K}a_{m}\left(\theta_{k},f_{k}\right)s_{k}\left(t\right)+n_{m}\left(t\right)\left(m=1,\ldots ,M\right)$$where $a_{m}\left(\theta_{k},f_{k}\right)=e^{-j2\pi \left(m-1\right)df_{k}\sin\left(\theta_{k}\right)/c}$ denotes the response entity of the *m*-th sensor with respect to the *k*-th (1≤k≤K) signal, c represents the speed of light, $s_{k}\left(t\right)$ represents the *k*-th incident far-field narrowband signal, $\theta_{k}$ and $f_{k}$ are the DOA and carrier frequency of the *k*-th signal. $n_{m}\left(t\right)$ denotes the noise signal of the *m*-th sensor. Then the received signal is sampled using a one-bit ADC, i.e.,
(2)$$x_{m}\left(t\right)=O\left(y_{m}\left(t\right)\right)$$where O(·) is the complex value quantization processing of the received signal, such as
(3)$$O\left(z\right)=\frac{1}{\sqrt{2}}\left(sign\left(real\left(z\right)\right)+jsign\left(imag\left(z\right)\right)\right)$$where if z > 0, sign(z) returns 1, otherwise returns −1. real(z) and imag(z) represent the real and imaginary parts of the complex value, respectively.

The relationship between the covariance matrix $R_{y}$ of $y_{m}\left(t\right)$ and the covariance matrix $R_{x}$ of the one-bit measurement $x_{m}\left(t\right)$ is discussed as follows. First of all, the *m*-th diagonal element of $R_{y}$ is $\delta_{y_{m}}^{2}$
(4)$$\delta_{y_{m}}^{2}={\left[R_{y}\right]}_{mm}=\sum_{k=1}^{K}{\left|a_{m}\left(\theta_{k},f_{k}\right)\right|}^{2}\delta_{k}^{2}+\delta_{n}^{2}$$where $\delta_{k}^{2}$ and $\delta_{n}^{2}$ represent the power of the *k*-th signal and noise, the element correlation coefficient of the (m,n)-th position of $R_{y}$ can be expressed as follows
(5)$$\begin{array}{l} \rho_{y_{m}y_{n}}=\frac{E\left[y_{m}\left(t\right)y_{n}^{*}\left(t\right)\right]}{\delta_{y_{m}}\delta_{y_{n}}}\\ =\frac{{\left[R_{y}\right]}_{mn}}{\sqrt{{\left[R_{y}\right]}_{mm}}\sqrt{{\left[R_{y}\right]}_{nn}}}\\ =\frac{\sum_{k=1}^{K}a_{m}\left(\theta_{k},f_{k}\right)a_{n}^{*}\left(\theta_{k},f_{k}\right)\delta_{k}^{2}}{\sqrt{\sum_{k=1}^{K}{\left|a_{m}\left(\theta_{k},f_{k}\right)\right|}^{2}\delta_{k}^{2}+\delta_{n}^{2}}\sqrt{\sum_{k=1}^{K}{\left|a_{n}\left(\theta_{k},f_{k}\right)\right|}^{2}\delta_{k}^{2}+\delta_{n}^{2}}}\\ =\frac{\sum_{k=1}^{K}a_{m}\left(\theta_{k},f_{k}\right)a_{n}^{*}\left(\theta_{k},f_{k}\right)\xi_{k}}{\sqrt{\sum_{k=1}^{K}{\left|a_{m}\left(\theta_{k},f_{k}\right)\right|}^{2}\xi_{k}+1}\sqrt{\sum_{k=1}^{K}{\left|a_{m}\left(\theta_{k},f_{k}\right)\right|}^{2}\xi_{k}+1}} \end{array}$$where $\xi_{k}$ is defined as
(6)$$\xi_{k}=\frac{\delta_{k}^{2}}{\delta_{n}^{2}}$$

Since all the sensors are identical, $\left|a_{m}\left(\theta_{k},f_{k}\right)\right|=1$. Therefore, Equation (4) can be simplified to
(7)$$\rho ={\left[R_{y}\right]}_{mm}=\sum_{k=1}^{K}\delta_{k}^{2}+\delta_{n}^{2}$$

Assuming that the powers of the signals are the same, i.e., $\xi_{k}=\xi$, then we have
(8)$$\rho_{y_{m}y_{n}}=\frac{\sum_{k=1}^{K}e^{j2\pi f\left(\tau_{m}\left(\theta_{i}\right)-\tau_{n}\left(\theta_{i}\right)\right)}}{K+\xi^{-1}}$$which implies $\left|\rho_{y_{m}y_{n}}\right|<1$. More importantly, we find the value of both the real and imaginary parts of the correlation coefficient decrease as ξ decreases.

Next, we focus on $R_{x}$. It is obviously that $x_{m}\left(t\right)$ is a zero-mean, unit variance, i.e., $E\left[x_{m}\left(t\right)\right]=0$, $\delta^{2}{}_{x_{m}}=1$. The (*m*,*n*)-th entity of $R_{x}$ can be expressed as
(9)$$\rho_{x_{m}x_{n}}=\frac{E\left[x_{m}\left(t\right)x_{n}^{*}\left(t\right)\right]}{\delta_{x_{m}}\delta_{x_{n}}}={\left[R_{x}\right]}_{mn}$$

It is obvious that if *m* = *n*, $\rho_{x_{m}x_{n}}={\left[R_{y}\right]}_{mm}=1$. According to the inverse sine law [37], we have
(10)$$\begin{array}{l} \rho_{x_{m}x_{n}}=\frac{2\arcsin e\left(\rho_{y_{m}y_{n}}\right)}{\pi }\\ ≜\frac{2}{\pi }\left(\arcsin\left(real\left\{\rho_{y_{m}y_{n}}\right\}\right)+j\arcsin\left(imag\left\{\rho_{y_{m}y_{n}}\right\}\right)\right) \end{array}$$

Hence the relationship between $R_{x}$ and $R_{y}$ is
(11)$$R_{x}=\frac{2}{\pi }\arcsin e\left(\frac{1}{a}R_{y}\right)$$

It can be seen from Equation (11) that
(12)Ry=asine(π2Rx)where sine(z)≜sin(real{z})+jsin(imag{z}), a is an unknown scaling factor. Furthermore, it is proven that [32]
(13)Rx≈2Ryaπ+(1−2π)Iwhere I is an identity matrix. Although a is unknown, we can see from Equation (13) that Rx and Ry share the same eigenvectors. This is why the traditional subspace-based algorithm can be directly applied to Rx for spectrum estimation. Now we consider L one-bit snapshots are available, and the quantified data is arranged into matrix format as
(14)X=O(AS˜T+N)where A=[a(θ1,f1),a(θ2,f2),⋯,a(θK,fK)]∈ℂM×K denotes the direction matrix, a(θk,fk)=[1,a2(θk,fk),⋯,aM(θk,fk)]T accounts for the steering vector, S˜∈ℂL×K is an unquantized signal measurement matrix, N is the array noise sample matrix. We know it that the signal subspace obtained from eigenvalue decomposition of the covariance matrix span the same subspace of the signal subspace achieved from singular value decomposition. Combined with the result of in Equations (13) and (14) can be rewritten as
(15)X0≈εAS˜TU+βNU=εAST+βE0where ε and β are scalers, U∈ℂL×L is a unitary matrix, S=UTS˜, E0=NU. Generally, K<min{M,P,L}. It should be pointed out that E0 is a Gaussian white noise matrix, as proven in [32].

## 3. The Proposed Framework

In this paper, the delay-based sampling framework is proposed for joint DOA and delayed estimation. As shown in Figure 1, the *P* delay units τp
(p=1,2,⋯,P) are follows the sensor array. If 0<τ1<τ2<⋯<1/max(fk), the array signal from the *p*-th delay units can be expressed as
(16)XP=εAΦPST+βEP,p=1,2,⋯,Pwhere(17)ΦP=[e−j2πf1τpe−j2πf2τp⋱e−j2πfKτp]∈ℂK×K

Define the delay matrix as
(18)Φ=[11⋯1e−j2πf1τ1e−j2πf2τ1⋯e−j2πfKτ1⋮⋮⋱⋮e−j2πf1τPe−j2πf2τP⋯e−j2πfKτP]∈ℂ(P+1)×K

Then Equation (16) defines a PARAFAC slicing model of the array output. In addition, outputs from all the delay units can be expressed as a third-order tensor χ with the (*m*,*l*,*p*)-th entity given by
(19)$$\begin{array}{l} \chi_{m,n,p}=\varepsilon \sum_{k=1}^{K}a_{m,k}{\tilde{s}}_{l,k}\phi_{q,k}+e_{m,n,q}\\ \left(m=1,2,\cdots ,M;l=1,2,\cdots ,L;q=1,2,\cdots ,P+1\right) \end{array}$$where $a_{m,f}$ is the (m,k)-th element of matrix A, ${\tilde{s}}_{l,k}$ is the (l,k)-th element of matrix S and $\phi_{q,k}$ is the ()-th element of matrix Φ. $e_{m,n,q}$ is the associate noise measurement.

The trilinear alternating least squares (TALS) algorithm is a very popular technique for trilinear models. The basic principle of TALS is to update one-factor matrix via least squares (LS) technique while treating other factor matrices as known parameters. Based on the previous estimation, TALS update the residual matrices successively. The above iterations will repeat until the converge conditions have been satisfied. The LS fitting for S is to solve
(20)$$\underset{A,S,\Phi }{\min}{\left\|\left[\begin{array}{c} X_{0}\\ X_{1}\\ \vdots \\ X_{P} \end{array}\right]-\left[\begin{array}{c} AD_{1}\left(\Phi \right)\\ AD_{2}\left(\Phi \right)\\ \vdots \\ AD_{P+1}\left(\Phi \right) \end{array}\right]S^{T}\right\|}_{F}$$where $X_{p}$
(p=1,2,⋯,P) is the *p*-th signal slice of χ from the ‘source’ direction. The LS update for S (denoted by $\hat{S}$) is then given by
(21)$${\hat{S}}^{T}={\left[\begin{array}{c} AD_{1}\left(\Phi \right)\\ AD_{2}\left(\Phi \right)\\ \vdots \\ AD_{P+1}\left(\Phi \right) \end{array}\right]}^{†}\left[\begin{array}{c} X_{0}\\ X_{1}\\ \vdots \\ X_{P} \end{array}\right]$$

Similarly, the *l*-th slice of χ from the ‘DOA’ direction can be expressed as $Y_{l}=\Phi D_{l}\left(S\right)A^{T}+{\;\tilde{N}}_{l}$, l=1,2,⋯,L, ${\tilde{N}}_{l}$ is the corresponding noise. The LS update for A (denoted by $\hat{A}$) is given by
(22)$${\hat{A}}^{T}={\left[\begin{array}{c} \Phi D_{1}\left(S\right)\\ \Phi D_{2}\left(S\right)\\ \vdots \\ \Phi D_{L}\left(S\right) \end{array}\right]}^{†}\left[\begin{array}{c} Y_{1}\\ Y_{2}\\ \vdots \\ Y_{L} \end{array}\right]$$

In addition, the slice of χ from the ‘frequency’ direction can be formulated as $Z_{m}\left(t\right)=SD_{m}\left(A\right)\Phi^{T}+{\bar{N}}_{m}$
(m=1,2,⋯,M), ${\bar{N}}_{m}$ is the corresponding noise. The LS update to Φ (denoted by $\hat{\Phi }$) is
(23)$${\hat{\Phi }}^{T}={\left[\begin{array}{c} SD_{1}\left(A\right)\\ SD_{2}\left(A\right)\\ \vdots \\ SD_{M}\left(A\right) \end{array}\right]}^{†}\left[\begin{array}{c} Z_{1}\\ Z_{2}\\ \vdots \\ Z_{M} \end{array}\right]$$

Before the first calculation of TLAS, A and Φ must be initialized. Usually, A and Φ are randomly initialized or initialized with ESPRIT or PM. Firstly, Equation (21) is computed to estimate S. Then, Equation (22) is calculated to update A (based on the initialized Φ and previously estimated S). Thereafter, Equation (23) is computed to update Φ. Finally, iterations in Equations (21)–(23) will repeat until convergence. In this paper, we adopted the COMFAC algorithm for PARAFAC decomposition [38], which can be quickly converge after only a few iteration.

It is well known to us that matrix decompositions are usually not unique unless some constrains are enforced. Unlike matrix decomposition, tensor decompositions are often unique under mild conditions. The following Theorem 1 gives the uniqueness of PARAFAC analysis.

**Theorem** **1.***Consider the matrices*A*,*Φ*, and*S*that establish the PARAFAC model in (19). If the k-rank of*A*,*Φ*,**and*S*(denoted by*kr(A)*,*kr(Φ)*, and*kr(S)*) satisfy*(24)kr(A)+kr(Φ)+kr(S)≥2K+2*Then the estimation of*A, Φ*, and*S*are unique up to permutation and scaling of columns, which can be expressed as*(25)$$\{\begin{array}{c} \hat{A}=A\Pi \Delta_{1}+E_{1}\\ \hat{\Phi }=\Phi \Pi \Delta_{2}+E_{2}\\ \hat{S}=S\Pi \Delta_{3}+E_{3} \end{array}$$*where*Π*is a permutation matrix,*$E_{1}$*,*$E_{2}$*, and*$E_{3}$*stand for the fitting errors, and*$\Delta_{1}$*,*$\Delta_{2}$*, and*$\Delta_{3}$*are diagonal scaling matrices with*$\Delta_{1}\Delta_{2}\Delta_{3}=I_{k}$*.*

The k-th column of the delay matrix Φ is
(26)$$g\left(f_{k}\right)={\left[1,e^{-j2\pi f_{k}\tau_{1}},\cdots ,e^{-j2\pi f_{k}\tau_{P}}\right]}^{T}$$

Thus, we can get the phase of $g\left(f_{k}\right)$ as
(27)$$h=-\mathrm{angle}\left(g\left(f_{k}\right)\right)={\left[0,2\pi f_{k}\tau_{1},\cdots ,2\pi f_{k}\tau_{P}\right]}^{T}$$

It is easy to find
(28)$$P_{1}b=h$$where(29)$$P_{1}=\left[\begin{array}{cc} 1 & 0\\ 1 & 2\pi \tau_{1}\\ \vdots & \vdots \\ 1 & 2\pi \tau_{P} \end{array}\right]$$where $b_{0}$ is a scaler that we do not care about. Suppose the k-th column of $\hat{\Phi }$ is $\hat{g}\left(f_{k}\right)$, and let the estimation of h is $\hat{h}$. The LS solution of b is
(30)$$\hat{b}=P_{1}^{†}\hat{h}$$

From the second entity of $\hat{b}$, we can get ${\hat{f}}_{k}$. On the other hand, we define
(31)$$\begin{array}{l} u=-angle\left(a\left(\theta_{k},f_{k}\right)\right)\\ ={\left[0,2\pi df_{k}\sin\theta_{k}/c,\cdots ,2\pi d\left(M-1\right)f_{k}\sin\theta_{k}/c\right]}^{T} \end{array}$$

Similarly, we have
(32)$$P_{2}c=u$$with(33)$$P_{2}=\left[\begin{array}{cc} 1 & 0\\ 1 & 2\pi df_{k}/c\\ \vdots & \vdots \\ 1 & \left(M-1\right)2\pi df_{k}/c \end{array}\right]$$where $c_{1}$ is a uninteresting constant. Replace $f_{k}$ with ${\hat{f}}_{k}$, we can get $P_{2}$. Let the estimation of u be $\hat{u}$, then the LS solution of c can be obtained via
(34)$$\hat{c}=P_{2}^{†}\hat{u}$$and finally the DOA can be estimated via
(35)$${\hat{\theta }}_{i}=\arcsin\left({\hat{e}}_{1}\right)$$

## 4. Simulation Results and Discussions

In this section, numerical simulations are carried out to verify the effectiveness of the proposed framework. In the simulations, we consider there are *K* = 3 uncorrelated far-field narrow-band signals with DOA-frequency pairs are $\left(10^{\circ },0.5\;\mathrm{MHz}\right)$, $\left(20^{\circ },0.7\;\mathrm{MHz}\right)$ and $\left(30^{\circ },0.9\;\mathrm{MHz}\right)$, and *L* snapshots are collected. An *M*-element ULA with half-wavelength spacing is adopted to receive the incoming signals. Assume there are 3 uniform delay units, with the delay interval is $10^{-7}$ s. The signal-to-noise ratio (SNR) is defined as $10\mathrm{lg}\left(P_{s}/P_{n}\right)$
(dB), where $P_{s}$ and $P_{n}$ are the powers with respect to signal and noise counterparts, respectively. The root mean square error (RMSE) is utilized for performance assessment. Herein, RMSE is defined as
(36)$$\mathrm{RMSE}=\sqrt{\frac{1}{Q}\sum_{q=1}^{Q}{\left(\vartheta_{q}-\vartheta_{0}\right)}^{2}}$$where $\vartheta_{q}$ is the estimated DOA or frequency of the *q*-th Monte Carlo trial, $\vartheta_{0}$ is the true value of the DOA or frequency, Q is the total number of Monte Carlo trials.

Firstly, we illustrate Q = 500 scatter results of the one-bit PARAFAC framework in Figure 2 and Figure 3, where M = 12 and L = 1000 are considered, and SNRs are set to 5 dB and 10 dB, respectively. It is shown that both DOA and frequency of the sources can be estimated and correctly paired from one-bit measurements. Moreover, it seems the estimated accuracy can be improved with increasing SNR, as the scatter results are more concentrated with larger SNR.

Secondly, the RMSE performance of the proposed algorithm with various SNR is depicted in Figure 4 and Figure 5, where M = 12 and L = 1000. For the performance comparison, the performance of traditional ESPRIT with one-bit measurement (marked with O-ESPRIT) and the traditional ESPRIT with un-quantified measurement (marked with U-ESPRIT) are added. RMSE with respect to DOA estimation and frequency estimation are shown in Figure 4 and Figure 5, respectively. It is seen that at low SNR regions (SNR < 0), there is no visible performance difference between the three algorithms. Notably, there is a performance gap between the algorithm with un-quantified measurements and that with one-bit measurement, this is because the one-bit measurement is lossy. As the SNR increases, the performance gap becomes larger. Another interesting finding is that when SNR is larger than a given threshold (SNR = 10), RMSE will not decrease with the increasing SNR. Besides, the proposed O-PARAFAC algorithm achieves better RMSE than O-ESPRIT, since the tensor structure has been exploited.

Thirdly, the RMSE curves with different snapshot number L are given in Figure 6 and Figure 7, where M = 12 and SNR = 0 dB are considered. Clearly, RMSE on DOA estimation and frequency estimation would improve with L increasing. Similarly, O-PARAFAC algorithm provides more accurate DOA and frequency estimation than O-ESPRIT. Also, both algorithms offer higher RMSE than the U-ESPRIT, since the one-bit measurement is lossy.

Fourthly, the RMSE performances versus sensor number M are depicted in Figure 8, Figure 9 and Figure 10, where L = 1000 and SNR = 0 dB. It is shown that estimation performance would improve with the increasing M. A similar observation can be seen that the performance corresponding to O-PARAFAC is between that of the O-ESPRIT and U-ESPRIT. In addition, one can observe that the proposed algorithm requires less calculation time then O-ESPRIT and U-ESPRIT when M is larger than 60, which implies that the proposed algorithms is much more efficient than the compared algorithms is the presence of massive antennas.

Finally, the RMSE curves with various delay P are depicted in Figure 11 and Figure 12, respectively, where M = 12 and SNR = 0 dB are considered. Similar to our previous findings, RMSE on DOA estimation and frequency estimation would improve with P increasing. In addition, the performance of O-PARAFAC is better than O-ESPRIT but worse than U-ESPRIT.

## 5. Conclusions

In this article, we stress the one-bit quantization problem in joint DOA and frequency estimation using a sensor array. A one-bit PARAFAC framework has been accomplished, in which the relationship of one-bit quantization and de-quantization measurement has been extended to tensor domain. Simulation results show the feasibility of the proposed framework. Since the one-bit quantization is lossy, the performance of the traditional subspace-based algorithms may degrade. However, the proposed one-bit PARAFAC can still offer good parameter estimation accuracy. As we all know, the one-bit quantization system is more flexible than the high-precision quantization and is less sensitive to storage capacity than the latter; the one-bit quantization system will have a bright prospect in further applications.

## Figures and Tables

**Figure 1 sensors-19-05422-f001:**
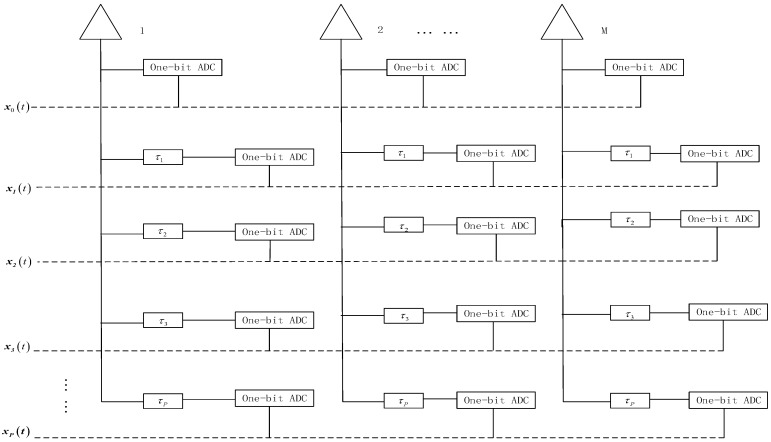
Architecture of the spatial-temporal sampling.

**Figure 2 sensors-19-05422-f002:**
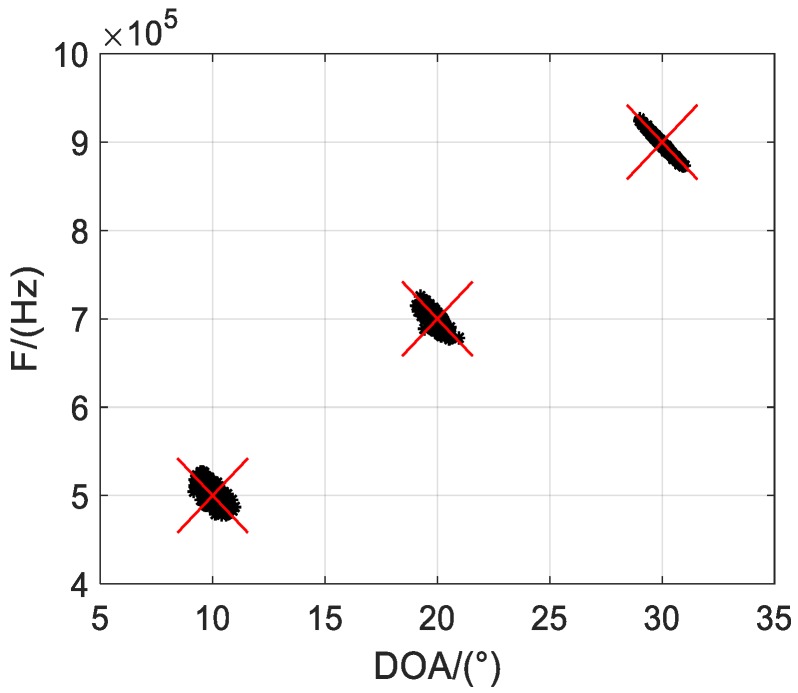
Scatter results with SNR = 5 dB.

**Figure 3 sensors-19-05422-f003:**
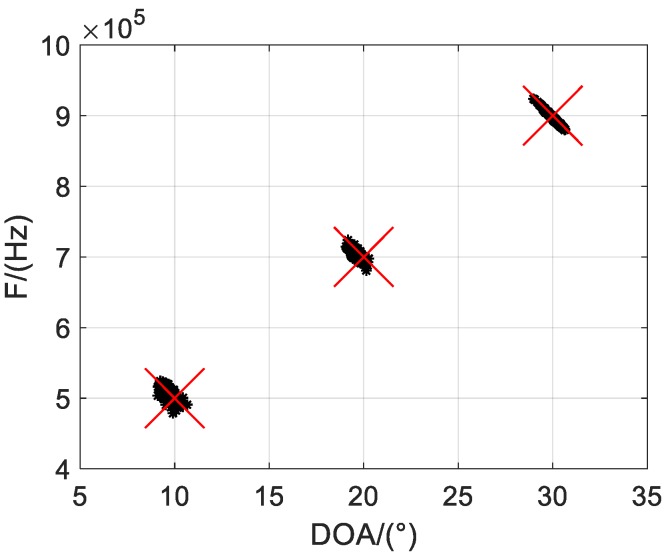
Scatter results with SNR = 10 dB.

**Figure 4 sensors-19-05422-f004:**
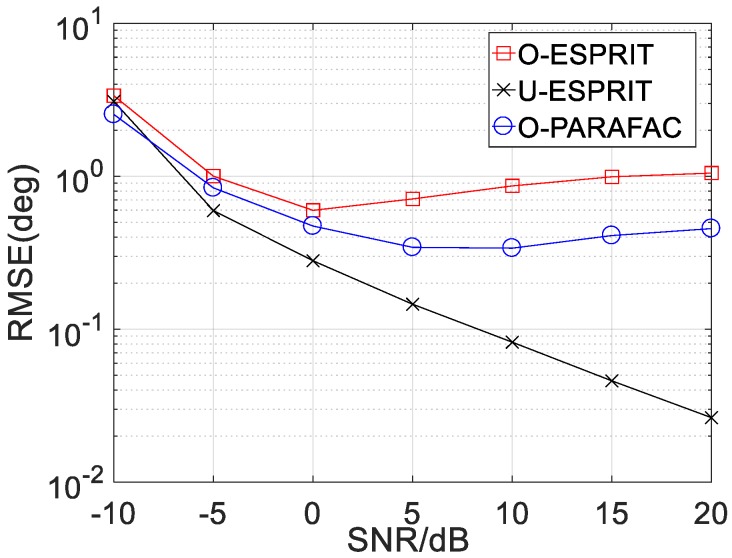
RMSE vs. SNR for DOA estimation.

**Figure 5 sensors-19-05422-f005:**
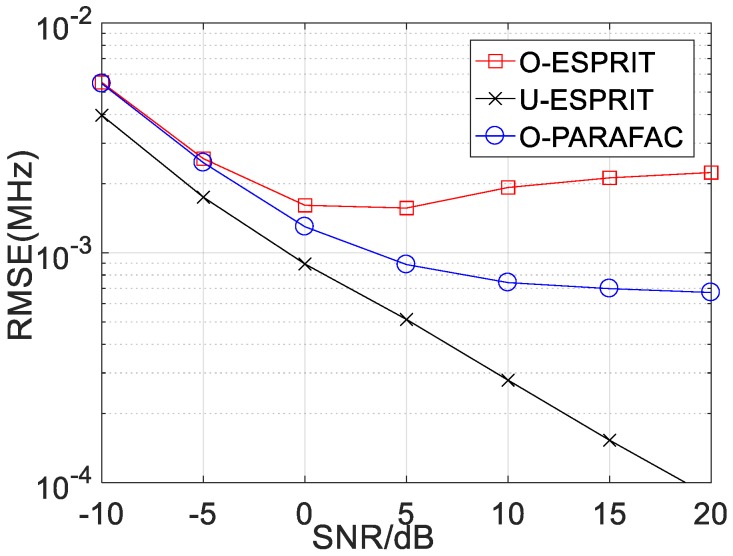
RMSE vs. SNR for frequency estimation.

**Figure 6 sensors-19-05422-f006:**
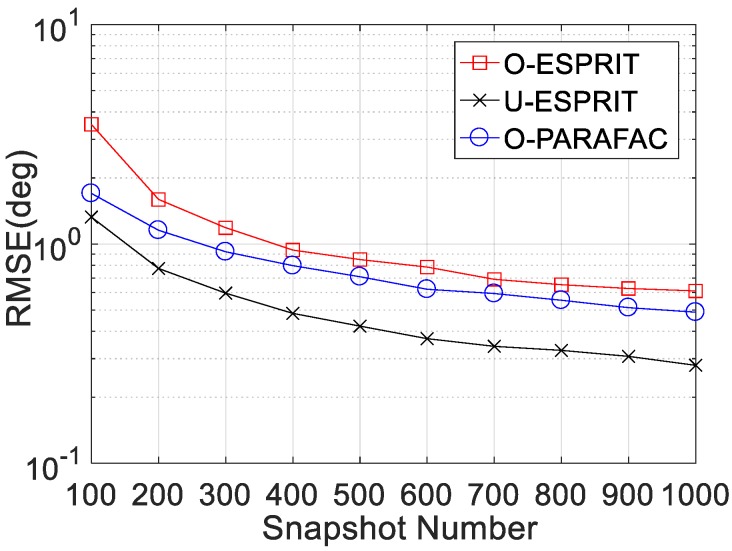
RMSE vs. L for DOA estimation.

**Figure 7 sensors-19-05422-f007:**
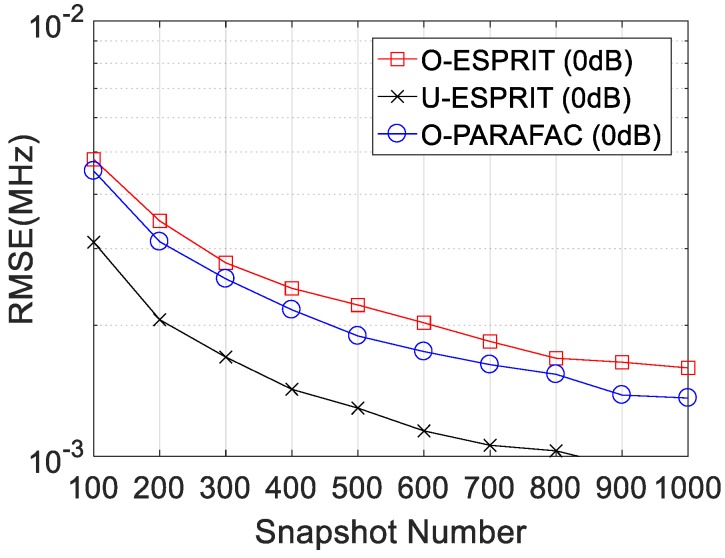
RMSE vs. L for frequency estimation.

**Figure 8 sensors-19-05422-f008:**
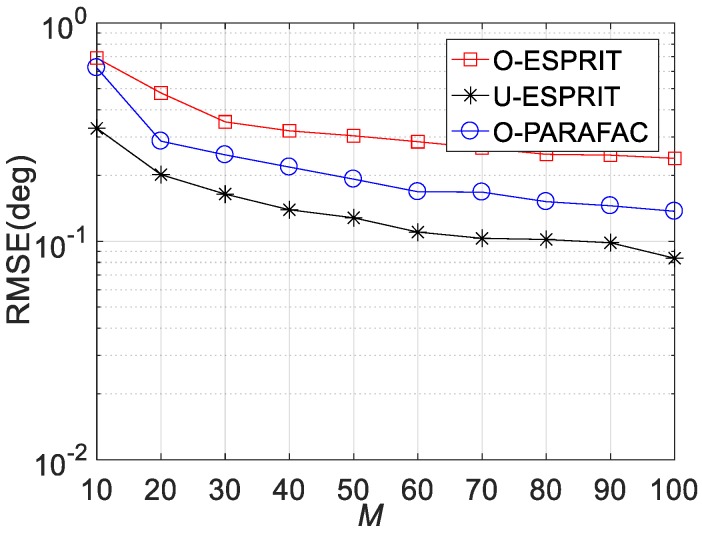
RMSE vs. M for DOA estimation.

**Figure 9 sensors-19-05422-f009:**
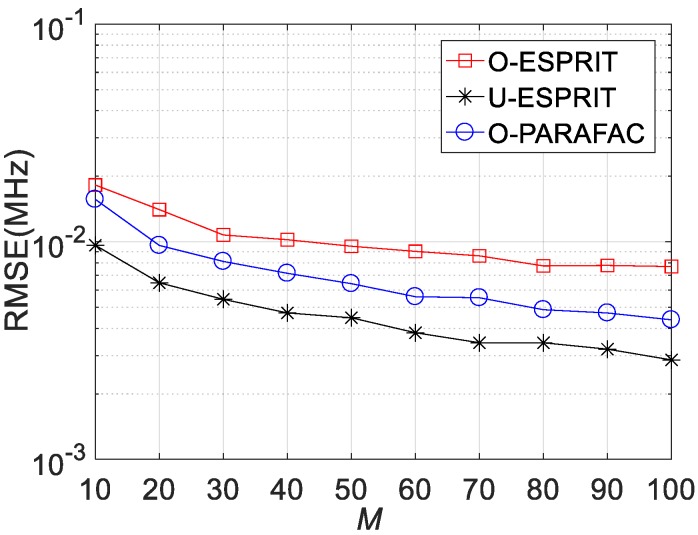
RMSE vs. M for frequency estimation.

**Figure 10 sensors-19-05422-f010:**
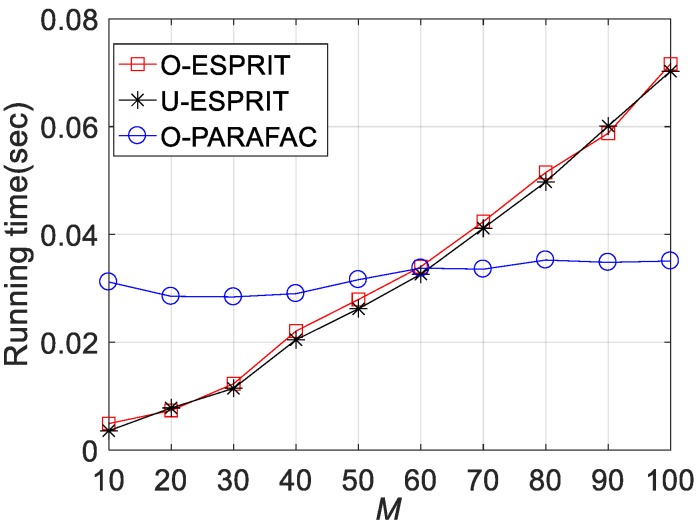
Average running time vs. M.

**Figure 11 sensors-19-05422-f011:**
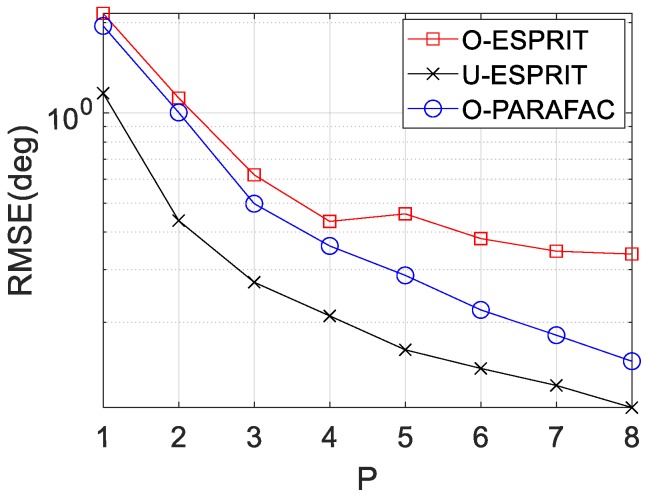
RMSE vs. P for DOA estimation.

**Figure 12 sensors-19-05422-f012:**
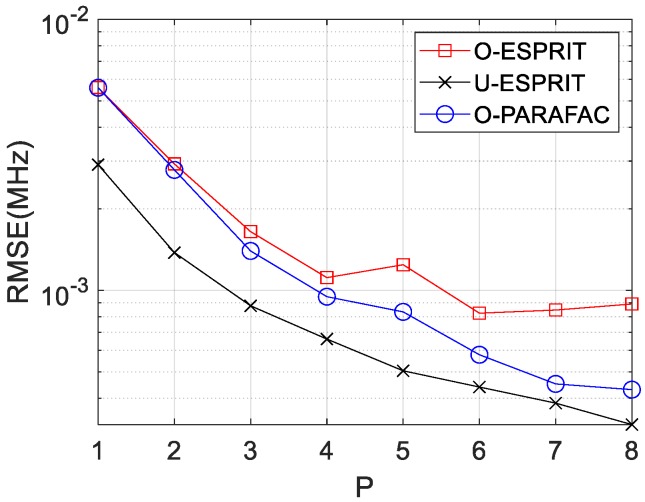
RMSE vs. P for frequency estimation.

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
