# Peer review of "Joint Angle and Frequency Estimation Using One-Bit Measurements"

_sensors, 2019, doi:10.3390/s19245422_

Round 1

Reviewer 1 Report

The logic of the paper is clear. here are some issue for the current version:

Authors should give more description on why joint angle-frequency estimation is important, why not joint angle-delay, joint delay-frequency or joint angle-frequency-delay? If my understanding is right, amk,fk) in equation (1) analogue to entry in the steering vector. However, it is better to explain more details of the properties of sk as the main purpose of the paper is to estimate the direction and frequency of the sk. In line 113, better to use "value" to replace "magnitude" It is better to clarify the meaning of operator ()T, as it is used in (14) (15).... without clarification It is better to clarify the meaning of operator ()†, as it is used in (21) (22).... without clarification Some symbols need to be redefined to avoid ambiguity. For example, \hat{A}, \hat{Φ} and \hat{S} are used in different lines (163, 167, 172) with different meaning. Better authors can find some ways to differentiate them. A similar situation happens to the symbol "k", at the beginning "k" is the index of the signal source, then, at line 179 "k" means the rank of the matrix.   Regarding the result, the reviewer expects to see the measurement of the computational benefits of the one-bit approach which is claimed at the beginning of the paper. 

Author Response

Comment1. Authors should give more description on why joint angle-frequency estimation is important, why not joint angle-delay, joint delay-frequency or joint angle-frequency-delay?

Response: Thank you for your instructive comments and suggestions. In this paper, we only focus on the problem of joint DOA and frequency estimation using a sensor array, as it is very important in various fields, such as in passive radar systems and wireless communications. Nevertheless, this is not mean other problems are less important. Actually, the problems of multiple parameter estimation using a sensor array are very similar to each other.

The revised Introduction section is given below (highlighted in blue color in the revision):

Page 1, line 28 to 36: Among the enormous research topics, joint angle and frequency estimation is so important in various fields that have been attached considerable interest. For instance, the two parameters can be adopted in passive radar systems for target locating and tracking; in space division multiple access-based wireless communications systems, these two parameters can be utilized to locate the user and allocate pilot tones; also, these two parameters are useful for channel estimation and beamforming. In this paper, we focus on the problem of joint DOA and frequency estimation with a sensor array. Nevertheless, this is not mean other multiple parameter estimation problems are less important. Actually, problems of multiple parameter estimation using a sensor array are very similar to each other.

Comment2. If my understanding is right, amk,fk) in equation (1) analogue to entry in the steering vector. However, it is better to explain more details of the properties of sk as the main purpose of the paper is to estimate the direction and frequency of the sk.

Response: Firstly, your understanding is correct and thanks for this comment. To explain the data model of (1) is really complicated, which involves the preliminaries of sensor array. In order to simplify this problem, the following reference contains the same data model of us is given in the revision. (see below)

[1]. Zhang X , Wang D , Xu D. Novel blind joint direction of arrival and frequency estimation for uniform linear array[J]. Prog. Electromagn. Res., 2008, 86: 199-215.

Comment3. In line 113, better to use "value" to replace "magnitude".

Response: Thank you for your kindly suggestion. According to your comments, we have used "value" to replace "magnitude" (see Line 125).

Comment4. It is better to clarify the meaning of operator ()T, as it is used in (14) (15).... without clarification; It is better to clarify the meaning of operator ()†, as it is used in (21) (22).... without clarification.

Response: Actually, we have given necessary definitions concerning all the used operators at the last paragraph of the Introduction section. The above mentioned symbols can be seen in Line 91.

Comment5. Some symbols need to be redefined to avoid ambiguity. For example, \hat{A}, \hat{Φ} and \hat{S} are used in different lines (163, 167, 172) with different meaning. Better authors can find some ways to differentiate them. A similar situation happens to the symbol "k", at the beginning "k" is the index of the signal source, then, at line 179 "k" means the rank of the matrix.

Response: Thank you for your instructive comments and suggestions. The previous expressions are misleading. Actually, \hat{A}, \hat{Φ} and \hat{S} are used in different lines have the same meaning. In order not to mislead the reader, we have

(a).updated the formulations in Eq. (21-(23). Also, necessary descriptions concerning TALS are added.

(b). redefined the k-rand operations.

The following revisions have been given in the revised manuscript.

Page 7, line 187 to 190: Before the first calculation of TLAS, A and Φ must be initialized. Usually, A and Φ are randomly initialized or initialized with ESPRIT or PM. Firstly, Eq. (21) is computed to estimate S . Then, Eq. (22) is calculated to update A (based on the initialized Φ and previously estimated S). Thereafter, Eq. (23) is computed to update Φ. Finally, iterations in Eq. (21)-(23) will repeat until convergence.

Page 8, line 195 to 196: If the k-rank of , and (denoted by,and) satisfy

                             kr(A) + kr(Φ) +  kr(S)  ≥2K+2                                ”

Comment6. Regarding the result, the reviewer expects to see the measurement of the computational benefits of the one-bit approach which is claimed at the beginning of the paper.

Response: Thanks for this helpful comment. According to your suggestion, we have added a new figure concerning average running time comparison (as shown in Fig 10.). In addition, necessary descriptions have been given.

Page 11, line 268 to 274: Fourthly, the RMSE performances versus sensor number M are depicted in Fig.8 - Fig.10, where L=1000 and SNR=0dB. It is shown that estimation performance would improve with the increasing M. A similar observation can be seen that the performance corresponding to O-PARAFAC is between that of the O-ESPRIT and U-ESPRIT. In addition, one can observe that the proposed algorithm require less calculation time then O-ESPRIT and U-ESPRIT when M is larger than 60, which implies that the proposed algorithms is much more efficient than the compared algorithms in the presence of massive antennas.

Reviewer 2 Report

This paper deals with joint DOA and frequency estimation using a sensor array with one-bit measurement, and a one-bit PARAFAC estimator is proposed. Compared with the closed-form subspace method, the proposed algorithm provides more accuracy estimation performance. Some comments for the authors' revision to improve the quality of this paper.

1. More literature review concerning one-bit quantization and signal processing with one-bit measurement should be given.

2. It is well known that PARAFAC algorithm perform well in Gaussian white noise, is the quantified noise is still Gaussian white?

3. I think the major problem of this paper is that it is unclear that how equation (15) get.

4. Some of the symbols have not been defined, e.g., ‘I’ in equation (13). I recommend the author carefully check the paper and made corresponding response.

5. ‘p’ in equation (20)-(21) should be ‘P’.

6. In equation (29), the symbol tau should not be in bold style.

7. Is there any bound parameter estimation?

8. How abuot the RMSE performance with various P?

Author Response

Comment1. More literature review concerning one-bit quantization and signal processing with one-bit measurement should be given.

Response: Thank you for your instructive comments and suggestions. According to the reviewer’s suggestion, more literatures are given.

Page 1, line 73 to 75: Besides, many efforts have been devoted to the sparse recovery problem from one-bit measurement [1-2]. In addition, some works have been done to the waveform design problems in MIMO radar with the problem of one-bit DAC [3].

Xiao P., Liao B., Li J., One-bit compressive sensing via Schur-concave function minimization, IEEE Trans. Signal Process., 2019, 67(16): 4139-4151. Xiao P., Liao B. Robust one-bit compressive sensing with weighted l1-norm minimization, Signal Process., 2019, 164: 380-385. Cheng Z., Liao B., He Z., et al. Transmit signal design for large-scale MIMO system with 1-bit DACs, IEEE Trans. Wirel. Commun., 2019, 18(9): 4466-4478.

Comment2. It is well known that PARAFAC algorithm perform well in Gaussian white noise, is the quantified noise is still Gaussian white?

Response: This is a very important question. It has been proven in ref. [32] that the noise is still Gaussian white after one-bit quantified.

Comment3. I think the major problem of this paper is that it is unclear that how equation (15) get.

Response: Thanks for this insightful comment. We think the expression in the previous version is not correct. The following revisions have been made in response to this comment.

Page 1, line 28 to 39: We know it that the signal subspace obtain from eigenvalue decomposition of the covariance matrix span the same subspace of the signal subspace achieved from singular value decomposition. Combined with the result of in Eq. (13), Eq (14) can be rewritten as

                              X0  ≈εAS^TUNU                                                                                             =εAS^TUE0

where ε and β are scalers,U is a unitary matrix, S=U^TS, E0=NU. Generally, K<min{M,P,L}. It should be pointed out that is a Gaussian white noise matrix, as proven in [32].

Comment4. Some of the symbols have not been defined, e.g., ‘I’ in equation (13). I recommend the author carefully check the paper and made corresponding response.

Response: Thank you for your instructive comments and suggestions. We carefully check the paper and redefine symbols that used in article.

Page 3, line 93 to 94:“ E[.] is to get the mathematical expectation of a variable. Dm(A) returns a diagonal matrix with the diagonal entities are the m-th row of A.

Page 5, line 139:where I is an identity matrix.

Comment5. ‘p’ in equation (20)-(21) should be ‘P’.

Response: We have updated the equations in the revised manuscript.

Comment6. In equation (29), the symbol tau should not be in bold style.

Response: So much thanks for this carefully comment. We have updated the symbols in the revision.

Comment7. Is there any bound parameter estimation?

Response: To the best of our knowledge, there only exists a lower bound on mean square error of parameter estimation with unquantified measurement. How to get the bound on parameter estimation with one-bit measurement is an interesting topic, but out the scope of this paper. We think we will pay attention to this question further.

Comment8. How about the RMSE performance with various P?

Response: Thank you for your suggestions. According to your suggestion, we test the RMSE performance with different P in the revision (see Fig11-12).

Page 12, line 275 to 280:” Finally, the RMSE curves with various delay P are depicted in Fig.11 and Fig.12, respectively, where M=12 and SNR= 0dB are considered. Similar to our pervious findings, RMSE on DOA estimation and frequency estimation would improve with P increasing. In addition, the performance of O-PARAFAC is better than O-ESPRIT but worse than U-ESPRIT.”

Round 2

Reviewer 1 Report

I think the authors still need a stronger justification for the choice joint angle-frequency estimation topic. The justification used in the current paper can be easily used for angle-delay, delay-frequency, angle-delay-frequency. I still cannot see what specific motivation or special reason why the angle-frequency estimation is chosen.  

I am fine with other responses. 

Author Response

Comment: I think the authors still need a stronger justification for the choice joint angle-frequency estimation topic. The justification used in the current paper can be easily used for angle-delay, delay-frequency, angle-delay-frequency. I still cannot see what specific motivation or special reason why the angle-frequency estimation is chosen.

Reply: We have re-stressed the motivation of why joint DOA and frequency estimation is chosen in this paper in the Introduction section, see the high-light words below:

"Source localization is one of the most important branches of array signal processing [1-2]. It has been actively conducted in the fields of communication, radar, sonar, seismic exploration and cognitive radio [3]-[7]. Source localization using a sensor array always involves spectrum estimation, such as direction-of-arrival (DOA) estimation, delay estimation, frequency estimation, polarization estimation, or a combination of them. Among the enormous research topics, joint angle and frequency estimation is particularly prominent, since the two parameters are very important in various fields, and they can improve the detection ability and anti-interference ability of the spatial source signals. For instance, the two parameters can be adopted in passive radar systems for target locating and tracking; in space division multiple access-based wireless communications systems, these two parameters can be utilized to locate the user and allocate pilot tones; also, these two parameters are useful for channel estimation and beamforming. Moreover, the algorithms using for joint DOA and frequency estimation can be easily extended for angle-delay estimation, delay-frequency estimation, angle-delay-frequency estimation, etc., since problems of multiple parameter estimation using a sensor array are very similar to each other. Due to the above reasons, we focus on the problem of joint DOA and frequency estimation in this paper."

Reviewer 2 Report

Good revision. No other comments.

Author Response

So many thanks for your help in improving this paper.